# Dual-Path Temporal Decoder for End-to-End Multi-Object Tracking

**Hyunseop Kim**[1][*]  **Juheon Jeong**[1][*]  **Hanul Kim**[2]  **Yeong Jun Koh**[1][†]

[1]Chungnam National University  [2]Seoul National University of Science and Technology

hyunseop95@gmail.com, jjh990427@gmail.com, hukim@seoultech.ac.kr, yjkoh@cnu.ac.kr

## Abstract

We present a novel end-to-end transformer-based framework for Multiple Object Tracking (MOT) that advances temporal modeling and identity preservation. Despite recent progress in transformer-based MOT, existing methods still struggle to maintain consistent object identities across frames, especially under occlusions, appearance changes, or detection failures. We propose a dual-path temporal decoder that explicitly separates appearance adaptation and identity preservation. The appearance-adaptive decoder dynamically updates query features using current frame information, while the identity-preserving decoder freezes query features and reuses historical sampling offsets to maintain long-term temporal consistency. To further enhance stability, we introduce a confidence-guided update suppression strategy that retains previously reliable features when predictions are unreliable. Extensive experiments on MOT benchmarks demonstrate that our approach achieves state-of-the-art performance across major tracking metrics, with significant gains in association accuracy and identity consistency. Our results demonstrate the importance of decoupling dynamic appearance modeling from static identity cues, and provide a scalable foundation for robust tracking in complex scenarios. Code is available at github.com/altkddhfcjs/DualTemporalMOT

## 1 Introduction

Multi-object tracking (MOT) aims to consistently estimate the spatial locations and identities of multiple objects across a video sequence. As a fundamental task in computer vision, MOT is essential for a broad range of real-world scenarios, including autonomous driving [35], surveillance [29, 43], sports analytics [9, 32], and crowd analysis [23], where consistent spatio-temporal tracking is required. Early MOT studies typically have followed the tracking-by-detection paradigm [1, 3, 17, 31, 42]. These methods first detect objects in each frame and then perform association steps across frames. These methods perform association steps based on the spatial proximity between objects in consecutive frames, measured by Interaction of Unions (IoU) [3, 6] or appearance similarity using ReID embeddings [1, 24, 30, 39]. These methods have benefited from rapid advances in object detection, but they remain limited in complex scenarios involving occlusion, similar appearance, and non-linear motions.

To address these limitations, recent approaches [12, 18, 36, 40] have adopted transformer-based MOT. These methods unify the DETR [7]-based detector and tracker in an end-to-end manner. In these models, track queries are propagated from the previous frame and used to associate and track objects over time. However, transformer-based MOT still faces a key challenge in maintaining the

---

[*]Equal contributions.

[†]Corresponding author.

39th Conference on Neural Information Processing Systems (NeurIPS 2025).

temporal consistency of track query features. Although these methods refine query features over multiple decoder layers, erroneous updates can accumulate and degrade feature consistency over time, ultimately resulting in identity switches.

In this work, we propose a transformer-based MOT framework with a novel dual-path temporal decoder that explicitly addresses this issue. Each decoder layer consists of two parallel components: an appearance-adaptive decoder layer that refines query features using current frame information and an identity-preserving decoder layer that maintains temporal consistency by reusing fixed query features and historical sampling offsets from the previous frame. The identity-preserving decoder layer reuses fixed query features and historical sampling offsets to enhance temporal stability and reduce sensitivity to abrupt changes in feature representation. To further enhance robustness, we introduce a confidence-guided update suppression strategy during inference, which retains the states of low-confidence track queries instead of updating them. This mechanism alleviates identity drift and improves tracking reliability in challenging scenarios such as occlusion and detection failure. Together, these designs enable the model to balance adaptability and stability, leading to more accurate and consistent multi-object tracking across long temporal spans.

The proposed MOT achieves the state-of-the-art performance on the DanceTrack [27] and SportsMOT [9] benchmarks, demonstrating strong association ability in challenging scenarios involving diverse objects with similar appearances.

In summary, the contributions are as follows:

- We propose a dual-path temporal decoder that disentangles appearance adaptation and identity preservation. The appearance-adaptive decoder layer dynamically refines query features using current-frame information, while the identity-preserving decoder layer keeps track queries fixed and reuses historical sampling offsets to maintain temporal consistency.
- We introduce a confidence-guided update suppression strategy that prevents unreliable updates under low-confidence conditions, thereby stabilizing identity association in the presence of occlusions and detection noise.
- Our method achieves new state-of-the-art performance on the DanceTrack [27] and SportsMOT [9] benchmarks, demonstrating strong improvements in both tracking accuracy and identity preservation.

## 2 Related Work

**Tracking-by-Detection.** Tracking-by-detection remains a dominant paradigm in MOT, where per-frame object detections are temporally associated to form object trajectories. SORT [3] employs the Kalman filter and the Hungarian algorithm to associate bounding boxes based on IoU. DeepSORT [31] enhances identity stability by combining the Kalman filter with appearance-based ReID embeddings. JDE [30], FairMOT [39], and Unicorn [33] jointly optimize detection and ReID features to learn discriminative representations and achieve consistent identity preservation. Recent efforts have extended this framework by improving detection quality and robustness against association errors. BoT-SORT [1] introduces appearance fusion, motion compensation, and the improved Kalman filter to strengthen real-time performance. Transformer-based approaches such as TransMOT [8] and GTR [42] model long-range spatiotemporal dependencies to enable more structured data association. OC-SORT [6] replaces heuristic motion models with learnable predictors for improved motion estimation. Extensions such as GHOST [26] and StrongSORT [10] focus on practical deployment by addressing domain shift, embedding refinement, and inference efficiency. More recently, DeconfuseTrack [16] formulates association as a multi-stage decision problem to reduce ID switches. DiffMOT [20] formulates data association as the denoising diffusion process to jointly predict object trajectories over time.

**Transformer-based MOT.** Recently, the transformer-based MOT has emerged as a promising direction in end-to-end MOT. Trackformer [22] and MOTR [36], both built upon on the DETR [7] architecture, perform joint detection and tracking by propagating track queries across frames within the decoder. MeMOT [5] extends this idea by incorporating both short- and long-term memories into track embeddings, improving robustness to occlusions and appearance changes. Subsequent works have explored improvements in temporal consistency, identity preservation, and scalability. MOTRv2 [40] enhances detection recalls by integrating external YOLOX [13] detections into the

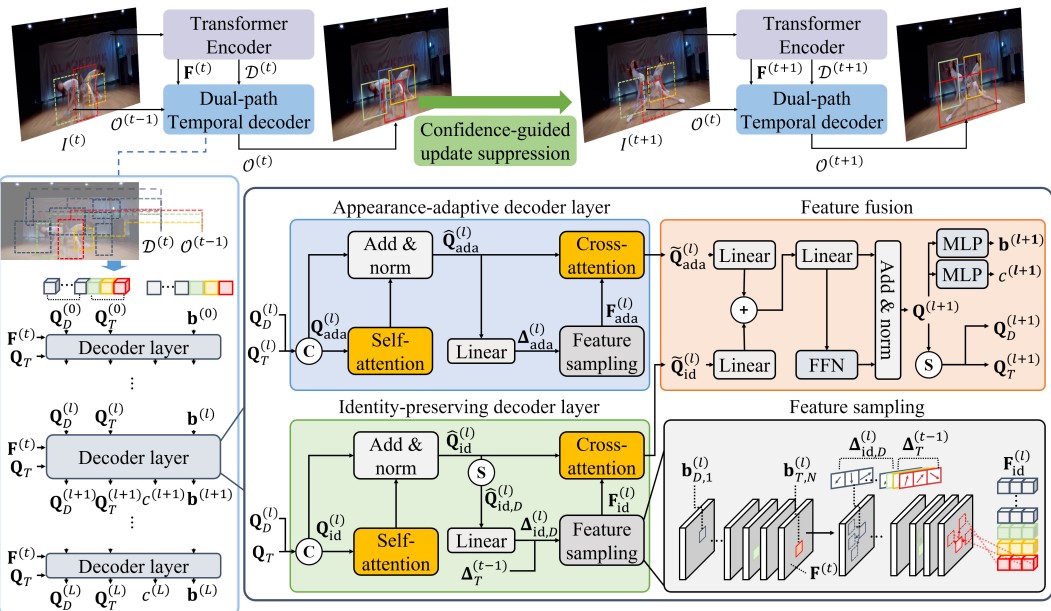

Figure 1: An overview of the proposed MOT framework. The dual-path temporal decoder consists of an appearance-adaptive decoder layer that refines query features using the current frame, and an identity-preserving decoder layer that maintains temporal consistency by freezing track query features and reusing historical sampling offsets. During inference, a confidence-guided update suppression strategy is applied to prevent unreliable feature updates. Ⓒ denotes concatenation and Ⓢ denotes feature splitting for track and candidate queries.

MOTR framework. MeMOTR [12] further strengthens identity stability under occlusion by directly injecting long-term memory into the transformer backbone. ColTrack [18] maintains identity consistency by applying self-attention between the current query and past queries along the same object's trajectory. MOTIP [11] explicitly decouples detection and association into two learnable modules, offering a more interpretable and flexible query interaction framework. However, these methods often suffer from inconsistent query updates, where inaccurate feature refinement across frames can degrade identity representations and lead to frequent identity switches.

## 3 Method

We aim to compose the set of tracked objects $\mathcal{O}^{(t)} = \{o_1^{(t)}, \ldots, o_N^{(t)}\}$ in each video frame $I^{(t)}$, where $N$ is the number of tracked objects. Each tracked object $o_n^{(t)}$ includes a bounding box $\mathbf{b}_{T,n}^{(t)} \in \mathbb{R}^4$, a confidence score $c_{T,n}^{(t)} \in \mathbb{R}^1$, a query feature $\mathbf{q}_{T,n}^{(t)} \in \mathbb{R}^C$, and sampling offsets $\mathbf{\Delta}_{T,n}^{(t)} \in \mathbb{R}^{K \times 2}$, where $C$ denotes the feature dimension and $K$ is the number of sampling points. For each frame $I^{(t)}$, we extract a feature map $\mathbf{F}^{(t)} \in \mathbb{R}^{H \times W \times C}$, where $H$ and $W$ denote height and width, and a set of $M$ detection candidates $\mathcal{D}^{(t)} = \{d_1^{(t)}, \ldots, d_M^{(t)}\}$ using a DINO [37]-based detector. Each detection candidate $d_m^{(t)}$ contains an anchor box $\mathbf{b}_{D,m}^{(t)} \in \mathbb{R}^4$ and a query feature $\mathbf{q}_{D,m}^{(t)} \in \mathbb{R}^C$. We aim to predict updated bounding boxes and confidence scores for tracked objects and detection candidates at the current frame $t$ based on thier previous states $\mathcal{O}^{(t-1)}$ and current observations $\mathcal{D}^{(t)}$.

Figure 1 illustrates the structure of the proposed MOT framework, which integrates a dual-path temporal decoder consisting of two complementary decoding layers. An appearance-adaptive decoder layer updates query features using the current frame, enhancing localization accuracy and adaptability to appearance changes. In parallel, an identity-preserving decoder layer maintains the original track queries and reuses historical sampling offsets, thereby preserving temporal consistency and alleviating identity drift. In addition, we introduce a confidence-guided update suppression strategy, which preserves previously reliable query features by preventing updates when current predictions are unreliable.

## 3.1 Dual-Path temporal decoder

The dual-path temporal decoder consists of stacked layers, each including two parallel branches: an appearance-adaptive decoder layer and an identity-preserving decoder layer. The appearance-adaptive decoder layer dynamically updates query features through deformable attention and sampling, while the identity-preserving path maintains temporal identity consistency by keeping track query features fixed and reusing their historical sampling offsets. For each decoder layer $l$, outputs from both branches are fused to refine the object representation and estimate bounding boxes and confidence scores.

**Appearance-adaptive decoder layer.** The appearance-adaptive decoder layers iteratively refine query features across $L$ decoder layers. For the initial decoder layer ($l = 0$), the track query features $\mathbf{Q}_T^{(0)} \in \mathbb{R}^{N \times C}$ are initialized from the propagated queries of the previous frame, where $n$-th row corresponds to a track query feature $\mathbf{q}_{T,n}^{(t-1)}$, while candidate query features $\mathbf{Q}_D^{(0)} \in \mathbb{R}^{M \times C}$ are initialized from the detection candidates, where $m$-th row corresponds to a candidate query feature $\mathbf{q}_{D,m}^{(t)}$. For subsequent decoder layers, both track and candidate query features, denoted as $\mathbf{Q}_T^{(l)}$ and $\mathbf{Q}_D^{(l)}$, respectively, are obtained as outputs from the previous decoder layer.

For each layer $l$, we concatenate these query features to form the combined query matrix $\mathbf{Q}_{\text{ada}}^{(l)} = [\mathbf{Q}_T^{(l)}; \mathbf{Q}_D^{(l)}]$. We then apply a multi-head self-attention, followed by residual connection and layer normalization to compute refined features

$$\hat{\mathbf{Q}}_{\text{ada}}^{(l)} = \text{LN}\left(\mathbf{Q}_{\text{ada}}^{(l)} + \text{SelfAttn}(\mathbf{Q}_{\text{ada}}^{(l)})\right). \tag{1}$$

We then predict the sampling offsets by applying a linear projection over the features $\hat{\mathbf{Q}}_{\text{ada}}^{(l)}$, *i.e.* $\boldsymbol{\Delta}_{\text{ada}}^{(l)} = \text{Linear}(\hat{\mathbf{Q}}_{\text{ada}}^{(l)}) \in \mathbb{R}^{(N+M) \times K \times 2}$, where $K$ is the number of sampling points. These offsets guide the deformable attention to extract features $\mathbf{F}_{\text{ada}}^{(l)} \in \mathbb{R}^{(N+M) \times K \times C}$ from the image feature map $\mathbf{F}^{(t)}$, inspired by the deformable attention mechanism in deformable DETR [44]. For the feature sampling process, the bounding box of the track query at the initial decoder layer $\mathbf{b}_{T,n}^{(0)}$ is initialized from the estimated bounding box at the previous frame, $\mathbf{b}_{T,n}^{(t-1)}$. Similarly, the bounding box of each candidate query $\mathbf{b}_{D,m}^{(0)}$ is initialized from the detection candidate's bounding box $\mathbf{b}_{D,m}^{(t)}$ at the current frame. Unlike the previous transformer-based MOT methods [18, 22, 40] that compute attention weights only from the query features, our approach aggregates the sampled image features $\mathbf{F}_{\text{ada}}^{(l)}$ based on affinity between query and sampled image features. Thus, a cross-attention layer is adopted to obtain the final enhanced query features $\mathbf{Q}_{\text{ada}}^{(l)}$ at layer $l$:

$$\tilde{\mathbf{Q}}_{\text{ada}}^{(l)} = \text{CrossAttn}\left(\hat{\mathbf{Q}}_{\text{ada}}^{(l)}, \mathbf{F}_{\text{ada}}^{(l)}\right) \tag{2}$$

where $\hat{\mathbf{Q}}_{\text{ada}}^{(l)}$ serves as query, while $\mathbf{F}_{\text{ada}}^{(l)}$ serves as key and value in the cross-attention.

**Identity-preserving decoder layer.** The identity-preserving decoder layer shares the same structure as the appearance-adaptive decoder, but handles track queries differently. By keeping track queries fixed across decoder layers and reusing historical sampling offsets, it prevents the injection of unstable evidence from the current frame, thereby preserving identity-specific features and ensuring temporal consistency across frames. Specifically, the track queries $\mathbf{Q}_T \in \mathbb{R}^{N \times C}$ are initialized from the propagated queries of the previous frame as in the adaptive decoder layer, but remain fixed across decoder layers to preserve temporal identity. For each layer $l$, the static track queries and candidate queries $\mathbf{Q}_D^{(l)} \in \mathbb{R}^{M \times C}$ are concatenated to form $\mathbf{Q}_{\text{id}}^{(l)} = [\mathbf{Q}_T; \mathbf{Q}_D^{(l)}]$. We then apply a multi-head self-attention, followed by residual connection and layer normalization to compute refined features

$$\hat{\mathbf{Q}}_{\text{id}}^{(l)} = \text{LN}\left(\mathbf{Q}_{\text{id}}^{(l)} + \text{SelfAttn}(\mathbf{Q}_{\text{id}}^{(l)})\right). \tag{3}$$

We split these refined features $\hat{\mathbf{Q}}_{\text{id}}^{(l)}$ into track and candidate components, denoted as $\hat{\mathbf{Q}}_{\text{id},T}^{(l)}$ and $\hat{\mathbf{Q}}_{\text{id},D}^{(l)}$, respectively.

Sampling offsets for candidate queries are obtained by applying a linear projection to $\hat{\mathbf{Q}}_{\mathrm{id},D}^{(l)}$, $i.e.$, $\boldsymbol{\Delta}_{\mathrm{id},D}^{(l)} = \mathrm{Linear}(\hat{\mathbf{Q}}_{\mathrm{id},D}^{(l)}) \in \mathbb{R}^{M \times K \times 2}$. In contrast, the offsets for track queries are reused from their historical offsets $\boldsymbol{\Delta}_T^{(t-1)}$ in the previous frame $t-1$ based on the observation that the same object typically exhibits similar sampling offset patterns across adjacent frames. Reusing historical offsets promotes spatial stability in attention, maintains alignment with persistent object regions, and suppresses transient noise arising from frame-specific variations. Although the track query features $\mathbf{Q}_T$ themselves do not change across decoder layers, their corresponding bounding boxes $\{\mathbf{b}_{T,1}^{(l)}, \ldots, \mathbf{b}_{T,N}^{(l)}\}$ are iteratively refined at each layer. Deformable attention uses the updated bounding box centers as reference positions, sampling image features $\mathbf{F}_{\mathrm{id}}^{(l)} \in \mathbb{R}^{(N+M) \times K \times C}$ from the image feature map $\mathbf{F}^{(t)}$ using $\boldsymbol{\Delta}_T^{(t-1)}$ and $\boldsymbol{\Delta}_{\mathrm{id},D}^{(l)}$. A cross-attention layer is then applied to obtain enhanced query features at layer $l$:

$$\tilde{\mathbf{Q}}_{\mathrm{id}}^{(l)} = \mathrm{CrossAttn}\left(\hat{\mathbf{Q}}_{\mathrm{id}}^{(l)}, \mathbf{F}_{\mathrm{id}}^{(l)}\right), \tag{4}$$

where queries are $\hat{\mathbf{Q}}_{\mathrm{id}}^{(l)}$, and keys and values are sampled features $\mathbf{F}_{\mathrm{id}}^{(l)}$.

**Feature fusion and state prediction.** For each decoder layer $l$, we fuse the outputs from the appearance-adaptive and identity-preserving decoder layers to obtain refined representations that incorporate both dynamic appearance changes and stable identity information. Specifically, given the enhanced query features from the appearance-adaptive path $\tilde{\mathbf{Q}}_{\mathrm{ada}}^{(l)}$ and from the identity-preserving path $\tilde{\mathbf{Q}}_{\mathrm{id}}^{(l)}$, we combine the corresponding track and candidate features from each path separately using linear projections, resulting in the fused track and candidate query features as follows:

$$\mathbf{Q}^{(l+1)} = \mathrm{Linear}\left(\mathrm{Linear}(\tilde{\mathbf{Q}}_{\mathrm{ada}}^{(l)}) + \mathrm{Linear}(\tilde{\mathbf{Q}}_{\mathrm{id}}^{(l)})\right). \tag{5}$$

The fused features are further refined using a feed-forward network (FFN) with a residual connection and layer normalization. Then, $\mathbf{Q}^{(l+1)}$ is split into $\tilde{\mathbf{Q}}_T^{(l+1)}$ and $\tilde{\mathbf{Q}}_D^{(l+1)}$, which are used for the next layer $l+1$. Here, $\tilde{\mathbf{Q}}_D^{(l+1)}$ is used for the appearance-adaptive and identity-preserving decoder layers, while $\tilde{\mathbf{Q}}_T^{(l+1)}$ is used for the appearance-adaptive layer only.

The refined fused query features $\mathbf{Q}^{(l+1)}$ are further used to predict bounding boxes and corresponding confidence scores for each object. Specifically, the bounding boxes $\mathbf{b}^{(l+1)} \in \mathbb{R}^{(N+M) \times 4}$ and confidence scores $c^{(l+1)} \in \mathbb{R}^{(N+M) \times 1}$ are obtained by passing $\mathbf{Q}^{(l+1)}$ through two separate multi-layer perceptrons (MLPs), each consisting of two linear layers with an activation function in between. These predictions serve as the updated object states at decoder layer $l+1$, facilitating accurate tracking and identification of objects across frames. The predicted bounding boxes and confidence scores are subsequently split into track and candidate components $(\mathbf{b}_T^{(l+1)}, c_T^{(l+1)})$ and $(\mathbf{b}_D^{(l+1)}, c_D^{(l+1)})$, respectively, for further sampling processing in subsequent decoding layers.

At the last layer $L$, the final tracking decisions for frame $t$ are determined based on confidence thresholding. Tracked objects whose predicted confidence scores $c_T^{(L)}$ are larger than a threshold $\alpha$ are regarded as active tracks, otherwise, they are marked as lost. Similarly, candidate objects with confidence scores $c_D^{(L)} > \alpha$ are determined as new tracks. Tracks that remain lost for more than $\tau$ consecutive frames are terminated. Finally, sampling offsets $\boldsymbol{\Delta}_T^{(t)}$ are extracted from the entries in $\Delta_{\mathrm{ada}}^{(L)}$, corresponding to both the track queries and the selected candidate queries, and are reused in the identity-preserving decoder at frame $t+1$.

## 3.2 Confidence-guided update suppression

While the confidence score determines whether an object is considered tracked or lost, conventional transformer-based tracking frameworks [18, 22, 36] continue to update query features of all track objects at every frame, regardless of their confidence. That is, even after an object is marked as lost, its query feature continues to be updated through self-attention and cross-attention. As a result, noisy observations from unreliable predictions are incorporated into the query representation. This can be problematic, since the updated query feature is subsequently used to predict the object's bounding box

and confidence score in the next frame. As a result, this unconditional update strategy accumulates noisy query features over time, which can cause identity drift or misassociation with nearby objects.

To address this issue, we propose a confidence-guided update suppression strategy. For objects with low confidence, we do not update their query features unless their predicted confidence exceeds a predefined threshold $\beta$. Instead, we preserve their previously reliable representations from earlier frames where their confidence was sufficiently high. This selective update mechanism suppresses the accumulation of noisy predictions from uncertain objects, thereby preserving clean identity embeddings and improving long-term tracking stability.

### 3.3  Training

We train our model in an end-to-end manner using a bipartite matching objective [7], as in previous transformer-based MOT frameworks [18, 36, 40]. Given the predicted tracked objects $\mathcal{O}^{(t)}$ and ground-truth set, we first compute the optimal assignment using Hungarian matching. The training loss is then computed over matched pairs using a weighted sum of a classification loss, a bounding box regression loss, and a generalized IoU loss, following standard practice.

## 4  Experiments

### 4.1  Datasets & Metrics

**DanceTrack [27]**. It is a multi-human tracking dataset in dancing scenes with similar uniform appearance and diverse motion, requiring strong association under occlusion and ambiguity. DanceTrack contains 40, 25, and 35 videos for training, validation, and test sets.

**SportsMOT [9]**. It is a recently released multi-object tracking dataset that focuses on athlete tracking in fast-paced sports such as soccer, basketball, and volleyball. The SportsMOT dataset presents significant challenges for motion modeling due to frequent acceleration and abrupt direction changes in these scenes. The dataset consists of 45, 45, and 150 sports sequences for training, validation, and test sets, respectively.

**MOT17 [23]**. It is a widely used pedestrian tracking dataset. MOT17 mainly contains massive pedestrians with simple and linear motions. It contains 7 training sequences and 7 test sequences. The sequences contain 500-1500 frames, recorded and annotated at 25-30 FPS.

**Metrics**. We assess the performance of the proposed method based on diverse MOT metrics for comparisons with other methods. Higher order tracking accuracy (HOTA) [19] is used as the primary metric, as it provides a balanced assessment of detection and association performance. To further dissect this trade-off, we report detection accuracy (DetA) and association accuracy (AssA), which decompose HOTA into its constituent factors. We also include ID F1 score (IDF1) [25], which measures the alignment between predicted and ground-truth identities, and multi-object tracking accuracy (MOTA) [2], a conventional metric that emphasizes detection errors, including false positives, false negatives, and ID switches.

### 4.2  Implementation Details

**MOT Network.**  The proposed framework is built on DINO [37] that uses ResNet-50 [14] backbone and transformer-based encoder. We select the top-$M = 300$ detection candidates from the encoder in DINO as anchor boxes and extract a candidate query feature $\mathbf{q}_{D,m}^{(t)}$ for each candidate by combining its learnable query embedding and positional embedding, following [37]. We set the number of dual-path temporal decoder layers to $L = 6$, a feature dimension to $C = 256$, and sampling points to $K = 256$. The confidence threshold $\alpha$, the suppression threshold $\beta$, and $\tau$ are set to 0.6, 0.4, and 60, respectively.

**Training.**  As in the prior works [18, 22, 36], we perform a two-stage training strategy. In the first stage, the object detector is trained for 40 epochs. In the second stage, the backbone and encoder are frozen, and only the dual-path temporal decoder is trained. The input images are resized to a resolution of $1440 \times 800$. The proposed MOT framework employs multi-scale training, Mosaic [4], and MixUp [15] for data augmentation. We use the AdamW optimizer with a learning rate of $1 \times 10^{-4}$

Table 1: Quantitative comparison on the DanceTrack [27] test set. The performance *with validation data* presents that the validation set is also included during training. The best results are boldfaced.

| Methods | HOTA | DetA | AssA | MOTA | IDF1 |
|---|---|---|---|---|---|
| *w/o valid data:* | | | | | |
| CenterTrack [41] | 41.8 | 78.1 | 22.6 | 86.8 | 35.7 |
| TransTrack [28] | 45.5 | 75.9 | 27.5 | 88.4 | 45.2 |
| ByteTrack [38] | 47.7 | 71.0 | 32.1 | 89.6 | 53.9 |
| QDTrack [24] | 54.2 | 80.1 | 36.8 | 87.7 | 50.4 |
| MOTR [36] | 54.2 | 73.5 | 40.2 | 79.7 | 51.5 |
| OC-SORT [6] | 55.1 | 80.3 | 38.3 | 92.0 | 54.6 |
| DiffMOT [20] | 62.3 | 82.5 | 47.2 | **92.8** | 63.0 |
| MeMOTR [12] | 68.5 | 80.5 | 58.4 | 89.9 | 71.2 |
| CO-MOT [34] | 69.4 | 82.1 | 58.9 | 91.2 | 71.9 |
| MOTRv2 [40] | 69.9 | 83.0 | 59.0 | 91.9 | 71.7 |
| MOTIP [11] | 72.0 | 81.8 | 63.5 | 91.9 | 76.8 |
| ColTrack [18] | 72.6 | - | 62.3 | 92.1 | 74.0 |
| Ours | **74.1** | **83.9** | **65.6** | 92.5 | **78.6** |
| *with valid data:* | | | | | |
| MOTRv2 [40] | 73.4 | 83.7 | 64.4 | 92.1 | 76.0 |
| ColTrack [18] | 75.3 | - | 66.9 | 92.2 | 77.3 |
| Ours | **76.2** | **85.0** | **68.3** | **92.5** | **79.9** |

Table 2: Quantitative comparison on the SportMOT [9] test set. The best results are boldfaced.

| Methods | HOTA | DetA | AssA | MOTA | IDF1 |
|---|---|---|---|---|---|
| QDTrack [24] | 60.4 | 77.5 | 47.2 | 90.1 | 62.3 |
| CenterTrack [41] | 62.3 | 82.1 | 48.0 | 90.8 | 60.0 |
| ByteTrack [38] | 62.8 | 77.1 | 51.2 | 94.1 | 69.8 |
| TrackFormer [22] | 63.3 | 66.0 | 61.1 | 74.1 | 72.4 |
| BoT-SORT [1] | 68.7 | 84.4 | 55.9 | **94.5** | 70.0 |
| MeMOTR [12] | 68.8 | 82.0 | 57.8 | 90.2 | 69.9 |
| TransTrack [28] | 68.9 | 82.7 | 57.5 | 92.6 | 71.5 |
| ColTrack [18] | 71.5 | 80.5 | 63.6 | 89.4 | 74.6 |
| OC-SORT [6] | 71.9 | **86.4** | 59.8 | **94.5** | 72.2 |
| DiffMOT [20] | 72.1 | 86.0 | 60.5 | **94.5** | 72.8 |
| MOTIP [11] | 72.6 | 83.5 | 63.2 | 92.4 | 77.1 |
| Ours | **73.9** | 82.2 | **66.6** | 91.5 | **78.7** |

and a weight decay of $1 \times 10^{-4}$. The learning rate is decayed by a factor of 0.1 during the final 15 training epochs. The model is trained for 45, 45 and 65 epochs on DanceTrack [27], SportsMOT [9] and MOT17 [23], respectively. All experiments are conducted on 8 NVIDIA RTX 4090 Ti GPUs with a batch size of 1, where each batch consists of a 4-frame video clip.

## 4.3 Benchmark Evaluation

**DanceTrack.** Table 1 shows the comparison of the proposed method with the existing methods on the test set in DanceTrack [27]. The proposed MOT achieves a HOTA score of 74.1 and achieves state-of-the-art performance across all metrics. In particular, compared to the previous best method ColTrack [18], it exhibits significant improvements in association accuracy, with AssA increasing from 62.3 to 65.6 and IDF1 increasing from 74.0 to 78.6. These results demonstrate the effectiveness of our temporal modeling in maintaining consistent object identities over time. Even when compared to prior works [18, 40] trained on both training and validation sets, our model consistently outperforms all competitors, demonstrating the effectiveness and robustness of our approach.

Table 3: Quantitative comparison on MOT17 [23] test set. The best results are boldfaced.

| Methods | HOTA | DetA | AssA | MOTA | IDF1 |
|---|---|---|---|---|---|
| *Heuristic:* | | | | | |
| OC-SORT [6] | 63.2 | - | 63.2 | 78.0 | 77.5 |
| ByteTrack [38] | 63.1 | 64.5 | 62.0 | 80.3 | 77.3 |
| BoT-SORT [1] | 64.6 | - | - | 80.6 | 79.5 |
| MixSort-OC [9] | 63.4 | 63.8 | 63.2 | 78.9 | 77.8 |
| MixSort-Byte [9] | 64.0 | 64.1 | 64.2 | 79.3 | 78.7 |
| Deep OC-SORT [21] | 64.9 | - | **65.9** | 79.4 | **80.6** |
| DeconfuseTrack [16] | **64.9** | **65.0** | 65.1 | **80.4** | **80.6** |
| *End-to-end:* | | | | | |
| MOTR [36] | 57.8 | 60.3 | 55.7 | 73.4 | 68.6 |
| MeMOTR [36] | 56.9 | 58.9 | 55.8 | 72.5 | 69.0 |
| TransTrack [28] | 54.1 | 61.6 | 47.9 | 74.5 | 63.9 |
| MOTRv2 [40] | **62.0** | **63.8** | 60.6 | 78.6 | 75.0 |
| TrackFormer [22] | - | - | - | 74.1 | 68.0 |
| ColTrack [18] | 61.0 | - | - | **78.8** | 73.9 |
| MOTIP [11] | 59.3 | 62.0 | 57.0 | 75.3 | 71.3 |
| Ours | 61.5 | 60.8 | **62.5** | 73.8 | **75.1** |

Table 4: Ablation studies for the identity-preserving decoder layer (IDL) on the DanceTrack [27] validation set. The best results are boldfaced.

| Method | HOTA | DetA | AssA | MOTA | IDF1 |
|---|---|---|---|---|---|
| without IDL | 66.7 | 76.1 | 56.3 | 87.0 | 69.5 |
| IDL with varying offsets $\mathbf{\Delta}_{\mathrm{id},T}^{(l)}$ | 67.5 | 77.5 | 59.0 | 87.1 | 73.2 |
| IDL with static historical offsets $\mathbf{\Delta}_{T}^{(t-1)}$ | **69.1** | **77.8** | **61.6** | **87.5** | **74.9** |

**SportsMOT.** Table 2 lists the performance on the SportMOT [32] test set. The proposed MOT achieves 73.9 HOTA, surpassing the previous state-of-the-art MOTIP [11] by margins of 1.3. Also, ours significantly improves the IDF1 score by 5.9 over DiffMOT [20], demonstrating superior association accuracy. DiffMOT, as a tracking-by-detection method reliant on pretrained detectors [13], excels on detection-centric metrics (*e.g.* MOTA, DetA) but underperforms on association-focused metrics such as IDF1.

**MOT17.** Table 3 presents the results on the MOT17 [23] test set. The proposed method achieves the best AssA and IDF1 performance among end-to-end approaches. It indicates that the proposed method maintains stable associations and is robust against ID switches. Despite our lower detection accuracy (MOTA 73.8) than ColTrack (MOTA 78.8) and MOTIP (MOTA 75.3), our stronger association capability enables higher HOTA and lower IDF1 than them, narrowing the gap to heuristic-augmented pipelines. Compared to MOTRv2, which uses heuristic post-processing, our method achieves superior AssA and comparable IDF1 while remaining fully end-to-end.

## 4.4 Ablation Study

We conduct ablation studies on the validation set of DanceTrack [27] to evaluate the effectiveness of the proposed components, including the identity-preserving decoder layer and the confidence-guided update suppression. In addition, We analyze the performance under various detection settings to further validate the robustness of the proposed framework.

**Identity-preserving decoder layer.** Table 4 reports the ablation study on the identity-preserving decoder layer (IDL). We evaluate three model variants to analyze its impact. First, we remove the IDL from the dual-path temporal decoder, reducing it to a single-path structure that consists of the appearance-adaptive decoder only. Second, we include the IDL but replace the historical sampling

Table 5: Ablation studies for the confidence-guided update suppression on DanceTrack [27] validation set. The best results are boldfaced.

| Method | $\beta$ | HOTA | DetA | AssA | MOTA | IDF1 |
|---|---|---|---|---|---|---|
| without confidence-guided update suppression | - | 67.9 | **78.1** | 59.2 | **87.6** | 72.9 |
| with confidence-guided update suppression | 0.2 | 68.2 | 77.6 | 60.2 | 87.3 | 73.8 |
| | 0.4 | **69.1** | 77.8 | **61.6** | 87.5 | **74.9** |
| | 0.6 | 68.7 | 77.9 | 60.7 | **87.6** | 74.2 |

Table 6: Comparison of the proposed method with other methods using various detectors on the DanceTrack [27] validation set. The best results are boldfaced.

| Detector | mAP | Tracker | HOTA | DetA | AssA | MOTA | IDF1 |
|---|---|---|---|---|---|---|---|
| YOLOX | 72.1 | MOTRv2 [40] | 64.5 | **78.7** | 53.0 | – | – |
| | | Ours | **67.9** | 77.2 | **60.0** | **87.2** | **73.3** |
| Deformable DETR | 63.7 | MOTIP [11] | 62.2 | 75.3 | 51.5 | 85.2 | 64.8 |
| | | Ours | **66.4** | **77.1** | **57.3** | **85.9** | **70.6** |
| DINO | 73.1 | ColTrack [18] | 61.9 | – | – | 86.5 | 61.6 |
| | | Ours | **69.1** | **77.8** | **61.6** | **87.5** | **74.9** |

offsets $\mathbf{\Delta}_T^{(t-1)}$ with predicted offsets $\mathbf{\Delta}_{\text{id},T}^{(l)}$, obtained by applying a linear projection to $\hat{\mathbf{Q}}_{\text{id},T}^{(l)}$. Finally, the full model includes the IDL with historical sampling offsets.

We observe that incorporating the identity-preserving decoder layer leads to consistent performance improvements over using only the appearance-adaptive decoder. Specifically, the variant with fixed track query features $\mathbf{Q}_T$ achieves notable gains, improving HOTA from 66.7 to 67.5 and IDF1 from 69.5 to 73.2. These results indicate that maintaining stable query features across decoder layers strengthens identity association and reduces drift. Furthermore, augmenting this with historical sampling offsets $\mathbf{\Delta}_T^{(t-1)}$ yields the best performance, achieving 69.1 HOTA and 74.9 IDF1. This underscores the importance of both feature consistency and temporally coherent attention for robust identity preservation over time.

**Confidence-guided update suppression.** Table 5 shows an ablation study to evaluate the impact of the confidence-guided update suppression strategy. Without this suppression, the model achieves 67.9 HOTA and 72.9 IDF1. In contrast, enabling the strategy consistently improves performance. Setting the threshold to $\beta = 0.4$ yields the best results, improving HOTA by 1.2 and IDF1 by 2.0, while also increasing AssA from 59.2 to 61.6. These results indicate that selectively retaining previously reliable query features under low-confidence conditions stabilizes identity association and reduces identity drift.

**Analysis under various object detectors.** To validate the generalization capability of the proposed dual-path decoder, we reproduced experiments on the DanceTrack validation using the same detectors adopted by prior methods. Table 6 shows the comparison of our model with previous transformer-based MOT under three detectors: YOLOX [13], Deformable DETR [44], and DINO [37], which are used in MOTRv2 [40], MOTIP [11], and ColTrack [18], respectively. The proposed method outperforms other transformer-based MOT methods for all detectors with significant HOTA improvements. These results demonstrate that the proposed dual-path decoder and stable query propagation consistently enhance association accuracy across all detector settings.

**Tracking results and sampling locations according to IDL.** Figure 2 illustrates a layer-wise comparison of tracking results and sampling locations from the appearance-adaptive decoder layer (ADL), with and without the identity-preserving decoder layer (IDL). For each setting, we visualize the predicted bounding boxes and the top 50 sampling points (based on attention weights) for multiple decoder layers ($l = 1, 2, 5, 6$) at the current frame $t$. Without IDL, the model fails to preserve the identity of the target object (green box) from frame $t - 1$, resulting in a misaligned bounding box

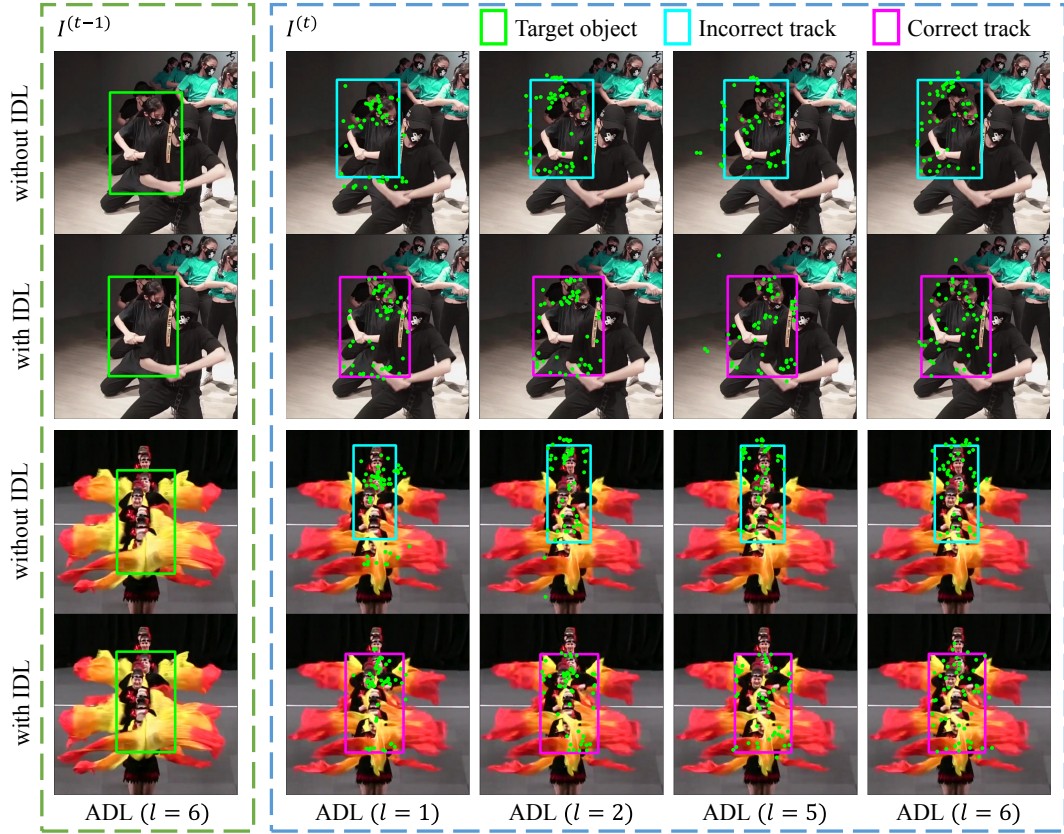

Figure 2: Visualization of tracking results and sampling locations in the appearance-adaptive decoder layer (ADL), with and without the identity-preserving decoder layer (IDL). The green box indicates the target object from frame $t - 1$. The cyan box represents an incorrectly tracked result, corresponding to an identity switch (IDSW), while the magenta box denotes the correctly tracked object. Green dots indicate the top 50 sampling locations with the highest attention weights.

at frame $t$ that corresponds to a different object (cyan box). The associated sampling points also shift toward this incorrect object, yielding a drift in query attention. In contrast, with IDL, the model consistently tracks the correct target object (magenta box) across layers, and the sampling points remain localized around the intended target. These observations demonstrate that IDL, by reusing static queries from the previous frame, enhances temporal stability and guides the decoder to maintain focus on the correct object during the iterative processes.

## 5 Conclusions

We introduced a transformer-based MOT framework that explicitly addresses the challenges of maintaining temporal consistency and robust identity association in complex scenes. At the core of our approach is a dual-path temporal decoder that decouples appearance adaptation from identity preservation, enabling the model to refine object representations while safeguarding identity-specific information from frame-specific noise. Additionally, we proposed a confidence-guided update suppression mechanism that further stabilizes tracking by selectively retaining reliable features under unreliable predictions. Through extensive experiments on DanceTrack and SportsMOT, our method consistently outperforms existing approaches in both detection and association metrics, establishing new state-of-the-art results on two benchmarks. The significant improvements in IDF1 and HOTA demonstrate the effectiveness of our temporal modeling strategy. We believe this work provides a strong foundation for further advancements in end-to-end multi-object tracking and highlights the importance of disentangling temporal dynamics and identity stability in transformer-based architectures.

**Acknowledgements.** This work was partly by the National Research Foundation of Korea (NRF) grants funded by the Korea government (MSIT) (No. RS-2024-00397293, No. RS-2024-00352566, No. RS-2025-00559165), and partly by Institute of Information & communications Technology Planning & Evaluation (IITP) grant funded by the Korea government (MSIT) (No.RS-2022-00155857, Artificial Intelligence Convergence Innovation Human Resources Development (Chungnam National University))

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
