# OpenReview forum: "Dual-Path Temporal Decoder for End-to-End Multi-Object Tracking"
_NeurIPS.cc/2025/Conference — NeurIPS 2025 poster_

### Official Review · Reviewer_sKej · 2025-06-24

**Clarity:** 3
**Significance:** 2
**Originality:** 2
**Rating:** 4
**Confidence:** 4

**Summary:**

This paper presents a transformer-based Multi-Object Tracking (MOT) framework with a dual-path temporal decoder that separates appearance adaptation and identity preservation. The key contribution is decomposing the temporal modeling into two parallel branches: an appearance-adaptive decoder that dynamically updates query features using current frame information, and an identity-preserving decoder that maintains temporal consistency by freezing track query features and reusing historical sampling offsets. Additionally, the paper introduces a confidence-guided update suppression strategy that prevents unreliable feature updates during inference. Experimental results demonstrate state-of-the-art performance on DanceTrack and SportsMOT benchmarks.

**Questions:**

1. Identity-Preserving Mechanism Validity: The reuse of sampling offsets $\Delta^{(t-1)}_T$ assumes similar attention patterns across consecutive frames. How does this approach handle scenarios with rapid object motion or camera movement where the optimal sampling locations should change significantly? Have you considered learning a dynamic offset adjustment rather than direct reuse?
2. Confidence-Guided Update Recovery: When an object has low confidence due to temporary occlusion or scene changes, the current mechanism freezes its features indefinitely until confidence recovers. What happens if the confidence threshold is never met again? Do you have mechanisms to prevent permanent feature degradation, and how do you handle re-emerging objects after extended periods?
3. Long-term vs. Short-term Trade-offs: The identity-preserving decoder only maintains consistency with the immediate previous frame ($t-1$). How does this limited temporal window perform in scenarios requiring longer-term memory? Have you experimented with incorporating features from multiple previous frames or implementing a more sophisticated temporal memory mechanism?
4. Notation Clarity and Experimental Fairness: The notation $Q_T$ and $Q_T^{(l)}$ can be confusing. Additionally, are the detector backbones consistent across all compared methods?

**Ethical Concerns:**

["NO or VERY MINOR ethics concerns only"]

**Final Justification:**

Overall, I still feel that the innovation aspect is lacking. However, the final experimental results are quite good. During the rebuttal stage, the discussion on innovation merely reiterates the main text without providing strong convincing power. Nonetheless, the related experiments do possess a certain degree of persuasiveness; therefore, I maintain my original score.

**Limitations:**

1. Temporal Window Limitations: The paper should acknowledge that the identity-preserving mechanism only considers $t-1$ frame information, which may be insufficient for complex long-term tracking scenarios.
2. Recovery Mechanisms: Lack of discussion about failure recovery when confidence-guided suppression leads to permanent feature freezing.

**Quality:**

3

**Strengths And Weaknesses:**

Strengths
1. Clear Technical Approach: The dual-path temporal decoder design is well-motivated and addresses a key challenge in transformer-based MOT, which maintains temporal consistency while adapting to appearance changes.

2. Strong Experimental Results: The method achieves state-of-the-art performance on challenging benchmarks (DanceTrack HOTA: 74.1, SportsMOT HOTA: 73.9).

3. Comprehensive Ablation Studies: The paper provides thorough ablation studies validating both the identity-preserving decoder layer and confidence-guided update suppression components.

Weaknesses
1. Limited Novelty in Individual Components: While the combination is effective, the individual components (deformable attention, confidence thresholding) are relatively standard. The core novelty lies primarily in the dual-path architecture design.
2. Questionable Identity-Preserving Strategy: The identity-preserving approach using $t-1$ frame offsets $\Delta^{(t-1)}_T$ seems overly simplistic for dynamic tracking scenarios. This only considers adjacent frame consistency rather than long-term temporal stability, which limits its effectiveness for long-range tracking.
3. Insufficient Confidence-Guided Update Suppression Justification: The strategy of completely discarding features below threshold $\beta$ may be too aggressive. In scenarios with sudden scene changes, objects might have consistently low confidence, leading to permanent feature freezing. The paper lacks a discussion of failure cases and recovery mechanisms.
4. Limited Long-term Tracking Analysis: The approach primarily addresses frame-to-frame stability but doesn't convincingly demonstrate improvements in long-term tracking scenarios with extended occlusions or re-identification.

---

> ### Author Rebuttal · Authors · 2025-07-31
>
> We sincerely appreciate the reviewer’s time and constructive feedback. Your insightful comments have helped us clarify and improve the manuscript, and we address each point in detail below.
>
> **A1.  Core novelty in the dual-path architecture design:**
>
> The comment notes that deformable attention and confidence thresholding are standard tools, but our contribution is not a simple combination of existing techniques. The dual-path temporal design is a novel mechanism that explicitly decouples adaptability and stability, a capability not addressed by prior transformer-based MOT approaches. We also introduce a modified affinity-based attention computation (lines 137–139) that enhances the dual-path decoder’s ability to align queries with relevant visual features, further improving association accuracy and identity preservation. Ablation results in Table 4 and the table below show that each of these contributions provides measurable gains, and their combination establishes a principled framework for temporal consistency control rather than a simple aggregation of standard components. We will include the new ablation study in the revised manuscript. Finally, our confidence-guided update suppression strategy selectively filters out unreliable updates under low-confidence predictions, ensuring that only stable and reliable query features are propagated to the next frame’s decoder.
> | **Attention weight computation method** | **HOTA** | **DetA** | **AssA** | **MOTA** | **IDF1** |
> |-----------------------------------------|----------|----------|----------|----------|----------|
> | Query-based                             | 67.6     | 77.6     | 59.1     | **87.7** | 72.7     |
> | Affinity-based (ours)                   | **69.1** | **77.8** | **61.6** | 87.5     | **74.9** |
>
>
> **A2. Validity of the identity-preserving mechanism under rapid motion:**
>
> The motivation behind re-using historical offsets is to mitigate feature drift caused by noisy updates, a known limitation of existing transformer-based MOT frameworks. This approach is designed based on the empirical observation (see Figs. S-1 and S-2 in the supplementary) that attention sampling patterns for the same object remain largely stable between consecutive frames. Our expectation is that by maintaining reliable associations between adjacent frames, we can achieve long-term temporal stability without requiring explicit long-horizon modeling, as temporal identity consistency is naturally preserved when consecutive links are robust.
>
> Regarding rapid object or camera motion, both paths in our dual-path decoder contribute to maintaining accuracy. The appearance-adaptive path dynamically updates queries to capture large displacements and sudden motion changes, while the identity-preserving path reuses stable offsets to prevent identity switches when motion patterns remain partially consistent. The table below shows the matching accuracy on the DanceTrack validation according to the frame-to-frame object displacement  for fast-motion analysis. The proposed combination with historical offset consistently improves the matching accuracy scores across all displacement ranges, including >40 px, indicating that our approach remains robust even in challenging fast-motion scenarios.
> | Distance (pixel) | without historical offset | with historical offset (Ours) |
> |------------------|---------------------------|--------------------------------|
> | 0 ~ 20          | 71.1                      | 73.4 (+2.3)                   |
> | 20 ~ 40         | 71.1                      | 73.1 (+2.0)                   |
> | 40 ~ ∞          | 63.0                      | 65.2 (+2.2)                   |
>
> We acknowledge that extreme conditions such as prolonged occlusions or abrupt camera changes may limit the effectiveness of direct offset reuse. Incorporating a learnable offset adjustment module, or combining offset memory with global motion cues, is a promising future direction we plan to explore. Nonetheless, current results demonstrate that our effective identity-preserving strategy already provides substantial gains in temporal stability without introducing additional complexity.
>
>
>
> **A3.  Effect of confidence-guided update suppression for consistent low confidence due to long-term occlusions:**
>
> The table below reports tracking accuracy on trajectories in DanceTrack validation that disappear from the ground truth for more than 30 frames due to occlusions and later reappear, evaluating how well each method recovers the correct identity after long-term absence, which yields consistent low confidence. Our method with confidence-guided update suppression achieves the highest score (0.192), outperforming that without suppression (0.188) and other transformer-based trackers such as MOTRv2 (0.139) and ColTrack (0.160). This demonstrates that discarding low-confidence updates does not harm re-identification and in fact improves robustness, as noisy updates are a stronger source of long-term identity drift.
>
> We acknowledge that in rare cases where confidence never recovers, a track may eventually fail due to outdated features not being refreshed. However, our analysis shows that the benefit of avoiding error propagation outweighs this risk, leading to higher overall recovery accuracy than competing methods. Long-term occlusion remains a challenging open problem for all trackers, as indicated by the generally low scores in the table below. We plan to address this limitation in future work by incorporating global memory mechanisms and learnable recovery strategies to better maintain identity information across long temporal gaps.
> | Method                                      | Long-term occlusion (> 30 frames) |
> |---------------------------------------------|------------------------------------|
> | MOTRv2                                      | 0.139                              |
> | ColTrack                                    | 0.160                              |
> | Ours without confidence-guided update suppression | 0.188                       |
> | Ours with confidence-guided update suppression    | **0.192**                      |
>
>
> **A4.  Long-term vs. short-term trade offs:**
>
> To further analyze long-term tracking, we divided trajectories into two groups based on their temporal span: short-term (≤1500 frames) and long-term (>1500 frames). For each trajectory, we computed the mean accuracy as the fraction of frames where the predicted bounding box overlaps the ground-truth with IoU ≥ 0.5, and report the average across all trajectories. As shown in the table below, identity-preserving decoder (IDL) improves accuracy from 68.7 to 70.3 (+1.6) in the short-term regime and from 58.3 to 60.1 (+1.8) in the long-term regime, demonstrating that IDL consistently enhances performance across both temporal scales by preventing error accumulation.
>
> Compared to other transformer-based trackers, our method achieves the highest accuracy in both regimes (70.3 / 60.1 vs. MOTRv2 65.2 / 58.1 and ColTrack 63.6 / 58.9), showing that the proposed dual-path design not only stabilizes adjacent-frame associations but also improves long-range identity preservation beyond existing approaches. We see that all trackers, including ours, exhibit reduced accuracy under long trajectories, highlighting that long-term tracking under extreme temporal gaps remains an open challenge. Future work will explore multi-frame memory modules or advanced temporal fusion mechanisms to further enhance re-identification robustness under extended occlusions or large spatial displacements.
>
> | Method              | Short-term (≤1500) | Long-term (>1500) |
> |--------------------- |------------------- |------------------ |
> | MOTRv2              | 65.2              | 58.1             |
> | ColTrack            | 63.6              | 58.9             |
> | Ours without IDL    | 68.7              | 58.3             |
> | Ours               | **70.3**          | **60.1**         |
>
>
> **A5.  Notation clarity and experimental fairness:**
>
> We acknowledge that the notations used in our paper may cause confusion. In the revised manuscript, we will clearly define each notation and improve readability. Regarding the question on experimental fairness, we confirm that all compared methods such as MOTR, TrackFormer, MOTRv2, ColTrack, and MOTIP use the same ResNet-50 detector backbone. We will explicitly state this in the revised manuscript to avoid any potential confusion.

---

> > ### Comment · Reviewer_sKej · 2025-08-04
> >
> > I sincerely appreciate the author's supplementary experiments demonstrating the algorithm's effectiveness. While I have some concerns regarding the overall innovation, I have ultimately decided to maintain my initial positive score, taking into account the coherence of the entire work and the persuasiveness of the supplementary experiments.

---

### Official Review · Reviewer_LCuU · 2025-07-01

**Clarity:** 2
**Significance:** 2
**Originality:** 3
**Rating:** 5
**Confidence:** 4

**Summary:**

The paper introduces a new multi-object tracking (MOT) architecture. The method is built upon DINO-DETR, a variant of the detection transformer (DETR).

The authors design **a new *dual-path* transformer decoder**. This decoder takes as input:
1. *track queries*: which come from objects detected at the previous frame
2. *candidate query features*: which are object candidates from the transformer encoder module (top-k scoring objects).

Every decoder layer is composed of two parallel decoder layers (i.e., the *appearance-adaptive* and the *identity-preserving* decoder layers) followed by a *feature fusion* module.
- The *appearance-adaptative* module iteratively refines the object queries using the image features, following DINO-DETR/Deformable-DETR.
- The *identity-preserving* module follows a similar design, but keeps track queries fixed (from one layer to the other) and reuses historical sampling offsets. This helps safeguard identity-specific information from the past.

The authors **additionally introduce a selective update mechanism** which only updates queries from previous frames if the object is predicted with high (classification) confidence, to help preserve clean identity embeddings and thus improve long-term tracking.

Experimentally, **the method outperforms previous works on the test sets from DanceTrack and SportsMOT**. It also achieves relatively high accuracy on MOT17's validation split.
Furthermore, the **ablation studies validate** the utility of the new dual path decoder and the confidence-guided update rule.

**Questions:**

1. Could the authors evaluate their method on the official test set from MOT17 instead of the validation set?
2. Could the authors discuss more in-depth the DETR-like related works (most importantly TransTrack, as it shares many similarities with this method)?
3. Do you observe any tracking performance degradation under fast motion? Indeed, one would expect that re-using historical offsets in the identity-preserving decoder layers may be detrimental in this case.

The reviewer is willing to raise the rating if the requests and questions are addressed; however, they will lower the rating if the two main concerns (1 and 2) remain unaddressed.


*[Remark: the reviewer is familiar with DETR-based MOT methods but is NOT an author of TransTrack]*

**Ethical Concerns:**

["NO or VERY MINOR ethics concerns only"]

**Final Justification:**

The other reviewers primarily raised similar concerns to mine, such as evaluating on the validation set rather than the test set (of MOT17) and potential challenges in scenarios involving fast motion. These issues have been adequately addressed. Additionally, the updated related works section now offers a clear comparison with the most relevant similar approaches.

Although the paper remains somewhat challenging to follow, the authors' clarifications and updates have significantly improved its quality. I thus believe the paper meets the required standard and accordingly raise my score to "accept".

**Limitations:**

No, the authors only briefly discuss limitations. The reviewer suggests discussing further the method's limitations and future research directions.

Furthermore, the paper does not discuss the potential negative societal impact of MOT (e.g., for autonomous killing robots, surveillance, etc.).

**Paper Formatting Concerns:**

/

**Quality:**

3

**Strengths And Weaknesses:**

### Strengths

- The method outperforms state-of-the-art methods on both DanceTrack and SportsMOT.
- The adaptive update rule is novel and could be beneficial in other methods as well.

### Weaknesses

- The Method section is not easy to read. It would benefit from being improved (e.g., by starting with an overview/intuition instead of notations, etc.).
- The related work section does not discuss the most similar work, TransTrack [1]. Discussing it would help readers understand how it is similar and how it differs from similar DETR-like approaches: The TransTrack paper, from 2020, also uses two types of queries, a dual transformer decoder, and confidence scores. However, this submission brings a new decoder design and introduces an interesting selective update mechanism.
- The paper is evaluated on the validation split from MOT17, rather than the official test set. This practice should be avoided, especially when the validation split comes from the same videos as the training split.


[1] Sun, Peize, et al. "Transtrack: Multiple object tracking with transformer." arXiv preprint arXiv:2012.15460 (2020).

---

> ### Author Rebuttal · Authors · 2025-07-31
>
> We sincerely appreciate the reviewer’s time and constructive feedback. Your insightful comments have helped us clarify and improve the manuscript, and we address each point in detail below.
>
> **A1. Enhancing clarity of the Method section:**
>
> We will revise the Method section by first providing an intuitive overview of the proposed model along, clearly illustrating the overall model architecture and tracking pipeline as shown in the example below..
>
> Figure 1 provides an overview of our proposed framework. Similar to prior DETR-based trackers, object queries are propagated across frames and updated by a transformer decoder. However, existing single-path designs struggle to maintain temporal consistency: noisy or low-confidence observations can corrupt query features, causing errors that accumulate over time and lead to identity switches. To address this, we introduce a dual-path temporal decoder that explicitly considers both adaptability and stability in query evolution. The appearance-adaptive path focuses on refining query embeddings with current-frame information for accurate localization and adaptation to visual changes. In parallel, the identity-preserving path retains stable features and reuses historical offsets, preventing feature drift when observations are unreliable. The outputs from both paths are fused, providing a balanced representation that is both flexible to genuine motion and robust to noise. A confidence-guided suppression mechanism further filters low-confidence updates, ensuring that only reliable information is propagated to subsequent frames. This architecture directly tackles the temporal stability problem that prior DETR-like methods leave unresolved, leading to improved long-term identity preservation and stronger association performance.
>
> **A2. Result on the MOT17 test set:**
>
> We were unable to include MOT17 test set results in the submission due to delays in account approval for test server access, preventing evaluation before the deadline. We have now completed this evaluation and report the results in the table below (with private detections). MOTRv2 &dagger; denotes the result of MOTRv2 without heuristic post-processing algorithms, as reported in the experimental results in the paper of MOTIP. Our method achieves HOTA 61.5, AssA 62.5, and IDF1 75.1, outperforming all non-heuristic end-to-end trackers, including MOTRv2† (HOTA 57.6, AssA 57.5, IDF1 70.3) and MOTIP (59.3 / 57.0 / 71.3). Despite lower detection accuracy (MOTA 73.8) than ColTrack (MOTA 78.8) and MOTIP (MOTA 75.3), our stronger association capability enables higher HOTA than all heuristic-free approaches, narrowing the gap to heuristic-augmented pipelines. Compared to the full MOTRv2 with heuristic post-processing, our method achieves superior AssA (62.5 vs. 60.6) and comparable IDF1 (75.1 vs. 75.0) while remaining fully end-to-end. We will include this table and explanation in the revised manuscript to clarify that our dual-path design achieves state-of-the-art association accuracy and competitive overall HOTA among purely end-to-end trackers, demonstrating its effectiveness in maintaining identities even with lower raw detection quality.
>
> | **Method**        | **HOTA** | **DetA** | **AssA** | **MOTA** | **IDF1** |
> |--------------------|----------|----------|----------|----------|----------|
> | **heuristic**      |          |          |          |          |          |
> | MixSort-OC         | 63.4     | 63.8     | 63.2     | 78.9     | 77.8     |
> | MixSort-Byte       | 64.0     | 64.1     | 64.2     | 79.3     | 78.7     |
> | Deep OC-SORT       | **64.9**     | -        | **65.9**     | 79.4     | **80.6**     |
> | DeconfuseTrack     | **64.9**     | **65.0**     | 65.1     | **80.4**     | **80.6**     |
> | **end-to-end** |       |          |          |          |          |
> | MOTR              | 57.8     | 60.3     | 55.7     | 73.4     | 68.6     |
> | TransTrack        | 54.1     | 61.6     | 47.9     | 74.5     | 63.9     |
> | MOTRv2&dagger;            | 57.6     | 58.1    | 57.5     | 70.1    | 70.3     |
> | MOTRv2            | **62.0**     | **63.8**     | 60.6     | 78.6     | 75.0     |
> | TrackFormer       | -        | -        | -        | 74.1     | 68.0     |
> | MOTIP             | 59.3     | 62.0     | 57.0     | 75.3     | 71.3     |
> | ColTrack          | 61.0     | -        | -        | **78.8**     | 73.9     |
> | **Ours**          | 61.5| 60.8 | **62.5** | 73.8| **75.1** |
>
>
> **A3.Comparison and discussion of DETR-like related works:**
>
> We agree that clarifying the relationship to DETR-like trackers, particularly TransTrack, will strengthen the paper. In the revised manuscript, we will expand Section 2 to highlight the architectural differences:
>
> - TransTrack: Employs the dual-decoder design, where one decoder handles detection queries and the other handles tracking queries. The two branches operate largely independently, extracting features separately for detection and tracking, without explicit mechanisms for cross-path feature refinement or identity stability.
>
>
> - MOTRv2, ColTrack: Use the single-path decoder that processes detection and tracking jointly. While this design unifies the tasks, it lacks any temporal stability control. Features for active tracks are continuously updated in the same pathway, making them prone to drift and identity switches under noisy observations.
>
>
> - Our method: Introduces the dual-path temporal decoder, where both paths jointly handle detection and tracking, but each is specialized for complementary roles: 1) the appearance-adaptive path dynamically refines query features to improve localization and adapt to visual changes and 2) the identity-preserving path maintains stable historical features and reuses past sampling offsets to prevent drift.
>
> The two paths are fused, combining adaptability and stability in a principled manner that is absent in prior DETR-like approaches. This formulation makes our approach conceptually distinct from both TransTrack and one-path designs (MOTRv2, ColTrack). We explicitly introduce temporal consistency control as a missing component in prior DETR-based trackers, leading to stronger association accuracy and fewer ID switches. We will revise the related work section to make these distinctions clear.
>
>
> **A4. Analysis of tracking performance under fast motion:**
>
> We conducted an additional analysis on the DanceTrack validation set, grouping trajectories by frame-to-frame object displacement to evaluate whether re-using historical offsets impacts tracking under fast motion. As shown in the table below, our method achieves consistently higher accuracy scores than the baseline without historical offsets across all displacement ranges. The improvement remains clear even for fast motions (>40 pixel), indicating that the dual-path decoder with historical offset reuse maintains robust tracking accuracy under fast motion. This stability arises because the appearance-adaptive path dynamically refines features for localization, while the identity-preserving path exploits historical offsets to preserve identity consistency.
> | Distance (pixel) | without historical offset | with historical offset (Ours) |
> |------------------|---------------------------|--------------------------------|
> | 0 ~ 20          | 71.1                      | 73.4 (+2.3)                   |
> | 20 ~ 40         | 71.1                      | 73.1 (+2.0)                   |
> | 40 ~ ∞          | 63.0                      | 65.2 (+2.2)                   |
>
>
>
> **A5. Discussion of method limitations, future directions, and societal impact:**
>
> We will add a dedicated limitations and future directions section discussing handling long-term occlusions and scalability to dense scenes, outlining paths for improvement. Furthermore, we acknowledge the potential negative societal impacts of MOT, such as surveillance misuse or autonomous weapon deployment, and will include a broader impact statement addressing dual-use risks and the importance of ethical safeguards. These additions will strengthen the paper’s transparency and societal awareness.

---

> ### Comment · Reviewer_LCuU · 2025-08-04
>
> Thank you for your detailed response.
>
> The other reviewers primarily raised similar concerns to mine, such as evaluating on the validation set rather than the test set (of MOT17) and potential challenges in scenarios involving fast motion. These issues have been adequately addressed. Additionally, the updated related works section now offers a clear comparison with the most relevant similar approaches.
>
> Although the paper remains somewhat challenging to follow, the authors' clarifications and updates have significantly improved its quality. I thus believe the paper meets the required standard and accordingly raise my score to "accept".

---

### Official Review · Reviewer_7Vx5 · 2025-07-02

**Clarity:** 3
**Significance:** 2
**Originality:** 2
**Rating:** 3
**Confidence:** 4

**Summary:**

The paper introduces a dual-path temporal decoder in transformer-based MOT, which separates appearance adaptation and identity preservation using two sets of decoders. The evaluation on the Dancetrack and SportMOT datasets demonstrates the effectiveness of the proposed dual-path temporal decoder.

**Questions:**

1. The performance of the tracker adopting standard DETR should be ablated.
2. Introduce the loss functions for each decoder.
3. Provide results on the test sets of the MOT17 and MOT20 datasets.
4. Provide the computational complexity analysis.

**Ethical Concerns:**

["NO or VERY MINOR ethics concerns only"]

**Final Justification:**

I maintain the negative rating as the performance is not even as good as MOTRv2 in HOTA on the MOT17 dataset, and it requires a larger computation cost. Although this justification is very strict, the performance of HOTA and computation is key for the MOT task.

**Limitations:**

Yes.

**Quality:**

2

**Strengths And Weaknesses:**

Strength
1. The concept of separating appearance and motion is commonly employed in tracking by detection methods. It is interesting to propose such thoughts in the transformer-based methods.
2. The presentation is good. The writing is easy to follow, and the figures are clear.

Weakness
1. The comparisons are not fair.  Other transformer-based methods adopt standard DTER as the object detector, while the proposed tracker adopts a DINO-based detector.
2. The loss functions are missing, which makes the learning objective of each decoder unclear.
3. The benchmark results on the MOT17 and MOT20 datasets are based on the validation sets, which share the same video sequences as the training sets. Such settings make the results not convincing enough.
4. Since the paper introduces two decoders compared to one decoder in existing methods, the computational complexity should be included and compared with existing methods.

---

> ### Author Rebuttal · Authors · 2025-07-31
>
> We sincerely appreciate the reviewer’s time and constructive feedback. Your insightful comments have helped us clarify and improve the manuscript, and we address each point in detail below.
>
> **A1. Comparison with different object detectors:**
>
> As shown in Table 1 (DanceTrack) and Table 2 (SportsMOT) of our paper, our method consistently outperforms ColTrack even when both trackers use the same DINO detector, demonstrating that the observed improvements are not simply due to adopting a stronger detector. To further ensure fairness, we also reproduced experiments on the DanceTrack validation using the detectors employed by other transformer-based MOT methods  (YOLOX for MOTRv2 and Deformable DETR for MOTIP). As in the table below, our tracker achieves superior results under identical detection inputs. These results validate that our gains mainly originate from the proposed dual-path temporal design and stable query propagation, rather than from detector choice. We will clarify this evaluation protocol and explicitly report detector-wise ablation results in the revised manuscript.
> | Detector        | mAP  | Tracker    | HOTA | DetA | AssA | MOTA | IDF1 |
> |-----------------|------|------------|------|------|------|------|------|
> | YOLOX           | 72.1 | MOTRv2     | 64.5 | **78.7** | 53.0 |   -  |   -  |
> |                 |      | Ours       | **67.9** | 77.2 | **60.0** | 87.2 | 73.3 |
> | Deformable DETR | 63.7 | MOTIP      | 62.2 | 75.3 | 51.5 | 85.2 | 64.8 |
> |                 |      | Ours       | **66.4** | **77.1** | **57.3** | **85.9** | **70.6** |
> | DINO            | 73.1 | ColTrack   | 61.9 |   -  |   -  | 86.5 | 61.6 |
> |                 |      | Ours       | **69.1** | 77.8 | 61.6 | **87.5** | **74.9** |
>
> **A2. Loss function:**
>
> In lines 206–210, we clarify that our framework uses the standard bipartite matching loss [7] adopted in transformer-based MOT methods [16, 35, 40]. Specifically, the overall loss is a weighted sum of focal classification loss, L1 box regression loss, and generalized IoU loss, computed after the outputs of the appearance-adaptive and identity-preserving decoders are fused. Since both paths jointly contribute to the final predictions, the loss is applied only once to the fused output. We will explicitly include this formulation in the revised manuscript to clarify the learning objective.
>
> **A3. Results on the MOT17 test set:**
>
> We were unable to include MOT17 test set results in the submission due to delays in account approval for test server access, preventing evaluation before the deadline. We have now completed this evaluation and report the results in the table below (with private detections). MOTRv2 &dagger; denotes the result of MOTRv2 without heuristic post-processing algorithms, as reported in the experimental results in the paper of MOTIP. Our method achieves HOTA 61.5, AssA 62.5, and IDF1 75.1, outperforming all non-heuristic end-to-end trackers, including MOTRv2† (HOTA 57.6, AssA 57.5, IDF1 70.3) and MOTIP (59.3 / 57.0 / 71.3). Despite lower detection accuracy (MOTA 73.8) than ColTrack (MOTA 78.8) and MOTIP (MOTA 75.3), our stronger association capability enables higher HOTA than all heuristic-free approaches, narrowing the gap to heuristic-augmented pipelines. Compared to the full MOTRv2 with heuristic post-processing, our method achieves superior AssA (62.5 vs. 60.6) and comparable IDF1 (75.1 vs. 75.0) while remaining fully end-to-end. We will include this table and explanation in the revised manuscript to clarify that our dual-path design achieves state-of-the-art association accuracy and competitive overall HOTA among purely end-to-end trackers, demonstrating its effectiveness in maintaining identities even with lower raw detection quality.
>
> | **Method**        | **HOTA** | **DetA** | **AssA** | **MOTA** | **IDF1** |
> |--------------------|----------|----------|----------|----------|----------|
> | **heuristic**      |          |          |          |          |          |
> | MixSort-OC         | 63.4     | 63.8     | 63.2     | 78.9     | 77.8     |
> | MixSort-Byte       | 64.0     | 64.1     | 64.2     | 79.3     | 78.7     |
> | Deep OC-SORT       | **64.9**     | -        | **65.9**     | 79.4     | **80.6**     |
> | DeconfuseTrack     | **64.9**     | **65.0**     | 65.1     | **80.4**     | **80.6**     |
> | **end-to-end** |       |          |          |          |          |
> | MOTR              | 57.8     | 60.3     | 55.7     | 73.4     | 68.6     |
> | TransTrack        | 54.1     | 61.6     | 47.9     | 74.5     | 63.9     |
> | MOTRv2&dagger;            | 57.6     | 58.1    | 57.5     | 70.1    | 70.3     |
> | MOTRv2            | **62.0**     | **63.8**     | 60.6     | 78.6     | 75.0     |
> | TrackFormer       | -        | -        | -        | 74.1     | 68.0     |
> | MOTIP             | 59.3     | 62.0     | 57.0     | 75.3     | 71.3     |
> | ColTrack          | 61.0     | -        | -        | **78.8**     | 73.9     |
> | **Ours**          | 61.5| 60.8 | **62.5** | 73.8| **75.1** |
>
>
> **A4. Computational complexity of the proposed dual-decoder:**
>
> Computational complexity comparisons are already included in our supplementary material (Table S-1), reporting FLOPs, parameter counts, and FPS against representative transformer-based MOT baselines (MOTRv2, MOTIP, ColTrack). These results show that our model remains parameter-efficient (11.3M vs. 41.7M for MOTRv2, 58.9M for MOTIP) and achieves comparable or better runtime speeds.
>
> To further address this point, we have added an additional experiment (the table below) directly comparing our dual-path decoder to a single-path variant without IDL. The results show only a negligible overhead (620.8G → 623.0G FLOPs, 13.0 → 12.7 FPS). This is because the two decoders partially share parameters, and their attention computations are executed in parallel, minimizing extra cost despite the dual-path architecture. We will make these details explicit in the main paper to clarify the computational trade-offs of the dual-path design.
>
> | Method                       | FLOPs (G) | Params (M) | FPS  |
> |------------------------------|-----------|------------|------|
> | MOTRv2                       | 431.8     | 41.7       | 25.7 |
> | MOTIP                        | -         | 58.9       | 12.0 |
> | ColTrack                     | 600.9     | 15.8       | 11.3 |
> | Ours without IDL (single-path decoder) | 620.8     | 10.9       | 13.0 |
> | Ours (dual-path decoder)     | 623.0     | 11.3       | 12.7 |

---

> ### Comment · Reviewer_7Vx5 · 2025-08-06
>
> I maintain the negative rating as the performance is not even as good as MOTRv2 in HOTA on the MOT17 dataset, and it requires a larger computation cost. Although this justification is very strict, the performance of HOTA and computation is key for the MOT task.

---

### Official Review · Reviewer_N3nk · 2025-07-03

**Clarity:** 3
**Significance:** 3
**Originality:** 3
**Rating:** 5
**Confidence:** 4

**Summary:**

This paper proposes a novel dual-path temporal decoder to enhance temporal object consistency in transformer-based track-by-query MOT frameworks (e.g., MOTR, TrackFormer). The decoder is integrated into the DINO detector, differing from the Deformable DETR detector used in MOTR/TrackFormer. Specifically, the proposed decoder comprises two parallel branches: an appearance-adaptive branch that refines query features using current frame information, and an identity-preserving branch that maintains temporal consistency by reusing fixed query features and historical sampling offsets (from the deformable attention in DINO) from the previous frame. Additionally, a query feature update strategy is introduced where features are not updated in frames with low confidence scores.
Quantitative experiments demonstrate SOTA performance of 76.2 HOTA on the DanceTrack dataset, and competitive performance on the SportsMOT dataset and MOT17 validation set. Ablation studies on the DanceTrack validation set validate the effectiveness of the main designs.

**Questions:**

See the weaknesses section. Each point corresponds to a suggestion.

**Ethical Concerns:**

["NO or VERY MINOR ethics concerns only"]

**Final Justification:**

The authors' rebuttal addressed my primary concerns:
* Lack of evaluation on the MOT17 test set: The newly reported performance (Table in A1) on the MOT17 test set demonstrates significantly better association performance in terms of IDF1 and AssA metrics than that of other end-to-end methods.
* Impacts of base detector: The newly added experiments (Table in A2) with YOLOX and Deformable-DETR offer a fair comparison with MOTRv2 and MOTIP. These results show the performance superiority stems from the proposed method's ability to achieve obviously stronger association performance than the counterparts, which well supports the central claim of the paper.
* Lack of validation of affinity-based attention: The authors have added an ablation study explicitly analyzing this component.
* Minor presentational issues: The authors will streamline the method section (which currently contains redundant notations), and move visualizations/efficiency comparisons from supplementary material to the main text.

Furthermore, considering the stronger association performance of their proposed method compared to other end-to-end MOT approaches and the new SOTA results achieved on the challenging DanceTrack dataset, I have raised my rating to Accept.

**Quality:**

3

**Strengths And Weaknesses:**

Strengths
1. The paper is clearly-written.
2. The designed dual-path temporal decoder presents a well-motivated and novel approach to enhancing temporal consistency.
3. The proposed method achieves excellent performance on the DanceTrack and SportsMOT datasets.  Specifically, it achieves 76.2 HOTA on DanceTrack, establishing a new SOTA result on this benchmark. Furthermore, it demonstrates excellent association performance, highlighting its effectiveness in improving temporal consistency and reducing ID switches.

Weaknesses
1. The paper reports results for MOT17 on the validation set but omits test set results. The reason for not reporting test set results is not provided.
2. While MOTRv2 identifies limited detection performance as a key weakness in MOTR and addresses it with an additional detector, and MOTRv3 attributes this to optimization imbalance between detection and track queries, the proposed method achieves better detection performance than MOTRv2. It remains unclear whether this improvement stems primarily from DINO's stronger detection capability and the query denoising technique. Analyzing DINO's contribution in the ablation study would offer valuable insights.
3. The main text lacks qualitative results (e.g., visualizations of tracking outputs, failure cases) demonstrating the method's effectiveness in improving association performance and reducing identity switches, particularly on challenging sequences. Additionally, the method section is overly detailed. Streamlining this section could save space to incorporate visualizations currently in the supplementary material
4. The computational overhead introduced by the dual-path decoder and the overall inference speed of the framework are not reported in the main text.
5. The modification to attention weight computation in the deformable attention (L137-139) is not validated in the ablation study.

---

> ### Author Rebuttal · Authors · 2025-07-31
>
> We sincerely appreciate the reviewer’s time and constructive feedback. Your insightful comments have helped us clarify and improve the manuscript, and we address each point in detail below.
>
> **A1. Results on the MOT17 test set:**
>
> We were unable to include MOT17 test set results in the submission due to delays in account approval for test server access, preventing evaluation before the deadline. We have now completed this evaluation and report the results in the table below (with private detections). MOTRv2 &dagger; denotes the result of MOTRv2 without heuristic post-processing algorithms, as reported in the experimental results in the paper of MOTIP. Our method achieves HOTA 61.5, AssA 62.5, and IDF1 75.1, outperforming all non-heuristic end-to-end trackers, including MOTRv2† (HOTA 57.6, AssA 57.5, IDF1 70.3) and MOTIP (59.3 / 57.0 / 71.3). Despite lower detection accuracy (MOTA 73.8) than ColTrack (MOTA 78.8) and MOTIP (MOTA 75.3), our stronger association capability enables higher HOTA than all heuristic-free approaches, narrowing the gap to heuristic-augmented pipelines. Compared to the full MOTRv2 with heuristic post-processing, our method achieves superior AssA (62.5 vs. 60.6) and comparable IDF1 (75.1 vs. 75.0) while remaining fully end-to-end. We will include this table and explanation in the revised manuscript to clarify that our dual-path design achieves state-of-the-art association accuracy and competitive overall HOTA among purely end-to-end trackers, demonstrating its effectiveness in maintaining identities even with lower raw detection quality.
> | **Method**        | **HOTA** | **DetA** | **AssA** | **MOTA** | **IDF1** |
> |--------------------|----------|----------|----------|----------|----------|
> | **heuristic**      |          |          |          |          |          |
> | MixSort-OC         | 63.4     | 63.8     | 63.2     | 78.9     | 77.8     |
> | MixSort-Byte       | 64.0     | 64.1     | 64.2     | 79.3     | 78.7     |
> | Deep OC-SORT       | **64.9**     | -        | **65.9**     | 79.4     | **80.6**     |
> | DeconfuseTrack     | **64.9**     | **65.0**     | 65.1     | **80.4**     | **80.6**     |
> | **end-to-end** |       |          |          |          |          |
> | MOTR              | 57.8     | 60.3     | 55.7     | 73.4     | 68.6     |
> | TransTrack        | 54.1     | 61.6     | 47.9     | 74.5     | 63.9     |
> | MOTRv2&dagger;            | 57.6     | 58.1    | 57.5     | 70.1    | 70.3     |
> | MOTRv2            | **62.0**     | **63.8**     | 60.6     | 78.6     | 75.0     |
> | TrackFormer       | -        | -        | -        | 74.1     | 68.0     |
> | MOTIP             | 59.3     | 62.0     | 57.0     | 75.3     | 71.3     |
> | ColTrack          | 61.0     | -        | -        | **78.8**     | 73.9     |
> | **Ours**          | 61.5| 60.8 | **62.5** | 73.8| **75.1** |
>
>
> **A2. Detector contribution analysis:**
>
> The improvement of our framework over MOTRv2 is not solely due to using a stronger detector (DINO) or its query denoising mechanism, but primarily to our dual-path temporal modeling. To ensure a fair comparison, we reproduced experiments on the DanceTrack validation using the same detectors adopted by prior methods as in the table below:
>
> - YOLOX detector (mAP 72.1, used in MOTRv2): Our method achieves HOTA 67.9 vs. 64.5 (MOTRv2), despite identical detection inputs.
>
>
> - Deformable DETR (mAP 63.7, used in MOTIP): Our method improves HOTA 66.4 vs. 62.2 and IDF1 70.6 vs. 64.8 over MOTIP under the same detection quality.
>
>
> - DINO (mAP 73.1, used in ColTrack): While providing a stronger baseline, our framework still achieves HOTA 69.1 vs. 61.9  and IDF1 74.9 vs. 61.6 over ColTrack, showing consistent gains independent of the detector choice.
>
> These results demonstrate that detection quality alone does not explain our improvements. The dual-path decoder and stable query propagation consistently enhance association accuracy across all detector settings. We will clarify this experimental setup and add an explicit detector-wise ablation in the revised manuscript.
>
>
> | Detector        | mAP  | Tracker    | HOTA | DetA | AssA | MOTA | IDF1 |
> |-----------------|------|------------|------|------|------|------|------|
> | YOLOX           | 72.1 | MOTRv2     | 64.5 | **78.7** | 53.0 |   -  |   -  |
> |                 |      | Ours       | **67.9** | 77.2 | **60.0** | 87.2 | 73.3 |
> | Deformable DETR | 63.7 | MOTIP      | 62.2 | 75.3 | 51.5 | 85.2 | 64.8 |
> |                 |      | Ours       | **66.4** | **77.1** | **57.3** | **85.9** | **70.6** |
> | DINO            | 73.1 | ColTrack   | 61.9 |   -  |   -  | 86.5 | 61.6 |
> |                 |      | Ours       | **69.1** | 77.8 | 61.6 | **87.5** | **74.9** |
>
>
>
> **A3. Qualitative results in main paper:**
>
> We thank the reviewer for the valuable suggestion. Our supplementary material (Figs S-3 to S-5) provides strong qualitative evidence showing that our method significantly reduces identity switches and improves association stability, particularly under challenging occlusions. We will move these key visualizations into the main text to better complement the quantitative gains reported. Additionally, we will streamline Section 3 by condensing redundant derivations and shifting technical details to the supplementary, freeing space for qualitative results. These revisions will make the manuscript more concise, accessible, and visually convincing while preserving technical rigor.
>
>
> **A4. Computational complexity:**
>
> The supplementary material (Table S-1) reports detailed efficiency metrics: our method achieves 12.7 FPS on NVIDIA 4090Ti while using fewer parameters (11.3M) than other end-to-end baselines (e.g., ColTrack 15.8M, 11.3 FPS), with only a minor FLOPs increase. We will incorporate these results into the main text and clarify that the additional branch adds negligible computational cost relative to single-path transformers while delivering substantial association improvements.
>
> **A5. Ablation on deformable attention weight computation:**
>
> We thank the reviewer for this insightful comment. We conducted an additional ablation study validating  the effectiveness of the proposed affinity-based attention weight computation (lines 137–139). As shown in the table below, replacing standard query-based weights with our affinity formulation yields consistent gains across all metrics on DanceTrack: HOTA 69.1 vs. 67.6, AssA 61.6 vs. 59.1, and IDF1 74.9 vs. 72.7, while maintaining similar DetA and MOTA. This demonstrates that computing attention weights from affinity between query and sampled image features improves association accuracy and identity preservation. We will include these results in the revised manuscript to explicitly validate this design choice.
>
> | **Attention weight computation method** | **HOTA** | **DetA** | **AssA** | **MOTA** | **IDF1** |
> |-----------------------------------------|----------|----------|----------|----------|----------|
> | Query-based                             | 67.6     | 77.6     | 59.1     | **87.7** | 72.7     |
> | Affinity-based (ours)                   | **69.1** | **77.8** | **61.6** | 87.5     | **74.9** |

---

### Note · Authors · 2025-08-13

We sincerely thank the reviewers for their constructive feedback. Several reviewers highlighted that our dual-path temporal decoder is a well-motivated (Reviewers N3nk and sKej) and novel approach (Reviewers N3nk and LCuU) that explicitly decouples adaptability and stability and performs adaptive update, addressing a key challenge in transformer-based MOT. Our method consistently achieves the state-of-the-art on DanceTrack in terms of HOTA and IDF1 scores across various detectors, indicating the observed gains are driven by the dual-path design in the proposed decoder rather than the specific detector choice. This stability across different detection backbones supports the robustness of the proposed approach in diverse settings. Our method also outperforms prior approaches on SportsMOT in both HOTA and association metrics, further demonstrating its generalization capability to varied tracking scenarios.

Our HOTA (61.5) is slightly lower than MOTRv2 (62.0) on MOT17, but this gap is largely attributable to differences in detection setup: MOTRv2 uses YOLOX detection boxes and heuristic linking strategies, whereas our method remains fully end-to-end. When focusing on the association metric AssA, which directly evaluates identity maintenance, our method achieves 62.5, surpassing both MOTRv2 (60.6) and other end-to-end trackers. This indicates that our approach delivers the strongest identity preservation performance in the end-to-end trackers.

As outlined above, multiple reviewers recognized the novelty of our dual-path temporal decoder, its ability to improve association accuracy across varied detectors, its strong results on benchmarks including DanceTrack and SportsMOT, and the potential broader utility of our selective update mechanism. We would be grateful if these points could be taken into consideration during the decision-making process.

---

### Decision · Program_Chairs · 2025-09-17

**Decision:**

Accept (poster)

**Comment:**

The paper presents a novel dual-path temporal decoder for multi-object tracking, integrating appearance and motion cues with confidence-guided updates. Reviewers unanimously acknowledge the technical soundness of the approach and its strong empirical performance, particularly on DanceTrack and SportsMOT datasets, with additional MOT17 test set results validating association accuracy. The authors have effectively addressed all major reviewer concerns in the rebuttal, including method clarity, ablation studies, detector-specific contributions, qualitative visualisations, related work comparisons (TransTrack, MOTRv2, ColTrack), and discussion of limitations and societal impacts. While one reviewer remains cautious regarding computational overhead and relative HOTA improvements on MOT17, the consensus is that the paper meets the standards for publication. All critical experiments and results from the rebuttal should be included in the final version. Based on these updates, the paper is recommended for acceptance by the AC and has been approved by the SAC.